# Stiff Person Spectrum Disorders—An Update and Outlook on Clinical, Pathophysiological and Treatment Perspectives

**DOI:** 10.3390/biomedicines11092500

**Published:** 2023-09-10

**Authors:** Benjamin Vlad, Yujie Wang, Scott D. Newsome, Bettina Balint

**Affiliations:** 1Department of Neurology, University Hospital Zurich, 8091 Zurich, Switzerland; benjamin.vlad@usz.ch; 2Department of Neurology, University of Washington, Seattle, WA 98195, USA; 3Division of Neuroimmunology and Neurological Infections, Department of Neurology, Johns Hopkins University School of Medicine, Baltimore, MD 21287, USA; 4Faculty of Medicine, University of Zurich, 8091 Zurich, Switzerland

**Keywords:** stiff person spectrum disorders, stiff person syndrome, glutamic acid decarboxylase, autoimmunity, autoimmune movement disorders

## Abstract

Stiff person spectrum disorders (SPSD) are paradigm autoimmune movement disorders characterized by stiffness, spasms and hyperekplexia. Though rare, SPSD represent a not-to-miss diagnosis because of the associated disease burden and treatment implications. After decades as an enigmatic orphan disease, major advances in our understanding of the evolving spectrum of diseases have been made along with the identification of multiple associated autoantibodies. However, the most important recent developments relate to the recognition of a wider affection, beyond the classic core motor symptoms, and to further insights into immunomodulatory and symptomatic therapies. In this review, we summarize the recent literature on the clinical and paraclinical spectrum, current pathophysiological understanding, as well as current and possibly future therapeutic strategies.

## 1. Introduction

Hitherto a rather unknown orphan disease, stiff person syndrome (SPS) recently received much attention even from the general public after famous singer Celine Dion declared that she had been diagnosed with SPS [1].

Moersch and Woltman were the first to describe “stiff-man syndrome” as an enigmatic disease with unusual trunk and leg stiffness in the absence of classic neurological features like spasticity or parkinsonism [2]. Later, the co-occurrence with other neurological and autoimmune diseases, most notably type 1 diabetes mellitus (T1DM), led to the discovery of the association with antibodies against glutamic acid decarboxylase (GAD) [3]. These early associations were critical for helping forge the road of understanding and defining what we nowadays call stiff person spectrum disorders (SPSD) [4,5]. On this basis, it was hypothesized that SPSD are autoimmune diseases with disturbances of the inhibitory GABAergic pathways [6]. Subsequently, autoimmunity against other antigens in inhibitory GABAergic pathways, like amphiphysin (indicative of paraneoplastic SPSD mainly due to breast or lung cancer; [7,8]), or inhibitory glycinergic pathways (Glycine receptor (GlyR) alpha 1 subunit [9,10]) were identified. These three autoantibodies appear to be the most relevant biomarkers of SPSD autoimmunity, though there are also other, rarer antibodies such as anti-DPPX [11].

Though the exact pathophysiological mechanisms of the aforementioned autoantibodies remain enigmatic, they can help support an SPSD diagnosis within the expanding clinical spectrum. Here, we give an update on new insights into the clinical and paraclinical features of SPSD and recent advances with regard to treatment.

## 2. The Emerging Clinical Spectrum beyond the Motor Symptoms of SPSD

Core symptoms of SPSD include fluctuating muscle stiffness with superimposed spasms, often leading to a prominent gait disturbance and an exaggerated startle response (hyperekplexia). There is a wide range of clinical manifestations (Figure 1), traditionally described as classic SPS (in which the stiffness and spasms are mainly in the lower back and leg muscles), stiff limb syndrome (more distal, often asymmetric stiffness/spasms), stiff person plus syndrome (classic features combined with additional neurological anomalies such as cerebellar ataxia or brainstem findings), acquired hyperekplexia and the potentially fatal progressive encephalomyelitis with rigidity and myoclonus (PERM) [12]. Additionally, there are a variety of emerging aspects beyond the classical motor features (Figure 2) [13,14].

### 2.1. More than Meets the Eye: The Retina

Optical coherence tomography (OCT) is gaining popularity as a neurological tool, not only for assessing acute optic neuritis, but also as a marker for neurodegeneration in patients without apparent visual impairment [15]. Lambe and colleagues addressed the structural and functional changes of the afferent visual system in SPSD patients, given the high number of GABAergic neurons enriched in the retina [16]. Both the thickness of the ganglion cell + inner plexiform layer (GCIPL) and the inner nuclear layer (INL) were reduced in SPSD patients compared to healthy controls, and the high- and low-contrast visual acuity scores were significantly worse compared to controls. The thickness of the GCIPL and INL appeared to be associated with the number of body regions affected (thinner retinal layers = more body regions involved) and the titer of serum GAD65 antibodies (>14,537 IU/mL). The majority of these associations remained even when excluding patients with DM. While the overall number of patients was low and with the high frequency of concomitant T1DM which may have a direct impact on the retina, OCT could be a potentially useful tool in the diagnostic workup of SPSD to estimate the burden of disease.

### 2.2. Difficult to Swallow, Hard to Digest: The Gastrointestinal System

Another hitherto under-researched aspect seems to be the impairment of the gastrointestinal (GI) system. Again, a link between GABAergic neurons and the enteric nervous system makes GI symptoms plausible in SPSD [17]. Dumitrascu and colleagues reported two cases with gastrointestinal and urological sphincter dysfunction as early manifestation of SPSD, with urinary retention, constipation, abdominal cramps and tenesmus as key clinical features [18]. Both cases were associated with GAD65 antibodies in serum, but differed in response to symptomatic and immunomodulatory therapy. In case 1, sphincter dysfunction did not respond to benzodiazepines and intravenous immunoglobulin (IVIg) but showed great response to rituximab, whereas IVIg was the key to symptom control in case 2. More recently, in a study involving 211 patients with SPSD, 25% reported GI symptoms including dysphagia, constipation and nausea. Dysmotility testing in a symptomatic subgroup was abnormal in almost two thirds of the patients [19]. Patients suffering from DPPX autoimmunity in particular are another important example as they suffer from chronic diarrhea due to hyperexcitability of gut neurons [20].

### 2.3. Gather your Breath: The Respiratory System

A freshly published paper focused on respiratory symptoms in SPSD and provided a detailed report on the prevalence of respiratory symptoms with spasms [21]. Over one third of the cohort suffered from such respiratory symptoms with a relevant impact on the modified ranking scale scores. While demographic and comorbidities did not appear to influence the prevalence, the number of symptomatic body regions was associated with the occurrence of respiratory symptoms. Two of the patients with respiratory symptoms with spasms ultimately died from secondary respiratory complications—both had PERM. Prior to that, preliminary work examined spirometry in SPSD patients with dyspnea and showed a frequent occurrence of a restrictive spirometry pattern [22]. Since respiratory failure causes high distress in patients and can, in the worst case, have a fatal outcome, a pulmonological assessment should be considered as part of routine clinical care. In an acute crisis, intranasal application of midazolam could be of special interest because of its rapid onset of action [23]. On the other hand, however, special caution is required with high doses of benzodiazepines, as they may worsen respiratory function, especially if co-administered in an emergency setting with opioids for treatment of associated pain. 

### 2.4. Mind Matters—Psychiatric and Cognitive Aspects

Beyond the neurological spectrum, a large number of patients present with psychiatric symptoms that have a significant impact on quality of life. Anxiety (56%) and depression (45%) were reported as major psychiatric complaints in a systematic literature review analyzing 239 SPSD patients with regard to their psychiatric symptoms [24]. Yet, treatment of depression is difficult because drugs such as tricyclic antidepressants and serotonin-norepinephrine reuptake inhibitors are suspected of worsening SPSD symptoms [25,26]. 

Closely related to psychiatric symptoms, cognitive impairment plays an important role in SPSD. In a retrospective screening of over 200 patients, approximately a third were reporting cognitive complaints; 20 of those patients underwent cognitive testing [27]. Impairments in verbal learning and recall memory (70%), verbal fluency (50%), attention (40%) and processing speed (40%) were the most common complaints. Possibly confounding factors in the majority of patients in this study, reflecting real-life experience, involve treatment with benzodiazepines, muscle relaxants or depression. 

While GABAergic drugs are an important pillar in the symptomatic treatment of SPSD motor symptoms and can be used for anxiety treatment, cognition may be impacted in some cases over time with long-term benzodiazepine intake, particularly if benzodiazepines with a short half-life are used [28,29,30]. Although, there are conflicting reports on the type of benzodiazepine (short vs. long-acting) used and the long-term impact on cognitive function. Psychiatric state and cognition influence each other and may be affected by the use of symptomatic medication, making it difficult to strictly separate one from the other. Appropriate screening for such symptoms should therefore be part of SPSD care.

## 3. The Diagnosis—About Swallows, Summers and Antibodies

With the increasing awareness of clinicians and advances in diagnostics, the diagnosis of antibody-mediated neurological disorders and SPSD can now be made more often than ever before [31]. However, Flanagan and colleagues recently highlighted the other side of the coin: based on antibody findings, functional neurological and psychiatric disorders were initially misdiagnosed as autoimmune diseases [32]. Reasons include false-positive test results and misinterpretation of low titer-positive antibody findings. Caution is warranted with immunoblot or line blot antibody positivity in isolation and antibody detection in noncertified laboratories. Relevant for SPSD in particular is the issue of low titer-positive GAD65 antibodies. Chia and colleagues reported GAD65 antibody positivity (mostly low titer) in 60% of patients misdiagnosed as having SPSD [33]. Low-titer GAD65 antibodies can occur as a non-specific finding, even in the healthy population [34,35]. Another example are onconeural antibodies such as amphiphysin antibodies, especially if low titer. As line blots are known to be liable to give “false positive” results, positive test results here need confirmation by immunohistochemistry [36]. Similarly, neuronal surface antibodies such as GlyR may occur as “antibodies of unknown significance”. This illustrates that a respective disease is not defined by the antibody test alone, just like the proverbial swallow does not make a summer [37]. Accordingly, the determination of antibodies without reasonable clinical suspicion should be avoided and unexpected positive antibody results should be reviewed for phenotypic compatibility and biological plausibility [38].

Of paramount importance in this context is the analysis of the cerebrospinal fluid (CSF). Abnormalities in CSF in SPSD can vary widely depending on the antibody. A comprehensive review of previously published basic CSF findings underlines the heterogeneity of CSF results [39]. While DPPX antibody-related disorders typically feature an inflammatory CSF, the basic CSF parameters of patients with GABAAR and GlyR autoimmunity are frequently unremarkable [40]. GAD65 autoimmunity, on the other hand, presents with CSF-restricted oligoclonal bands in more than 50% of cases and GAD65 antibodies can be detected in many cases of SPSD [39,41]. Much less is known about the rarer forms of SPSD. Thus, evidence of CSF abnormalities reinforces the suspicion of SPSD, but the absence of abnormalities does not exclude it. In any case, when testing for antibodies, it is recommended to analyze both serum and CSF to avoid false-positive or false-negative findings and to achieve the highest possible sensitivity and specificity [37]. The most reliable method to detect a relevant antibody finding is the detection of intrathecal antibody-specific immunoglobulin synthesis, as it is a direct sign of highly specific central nervous system autoimmunity [42,43]. 

## 4. About Dogs, Horses, Mice and Men in SPSD

SPSD do not only affect humans. In 2000, the occurrence of stiffness and spasms in the axial muscles of the lower back and hind legs was reported for the first time in a horse [44]. Due to the similarity to the human phenotype of SPSD, this condition was named “stiff-horse syndrome”. Indeed, there were significant titers of GAD65 antibodies in the serum and a very good response to steroid treatment. Over the last few years, more cases of stiff-horse syndrome and a case of “stiff-dog syndrome” have been reported, all associated with GAD65 antibodies and a good response to steroid treatment [45,46,47].

On the other hand, however, there is still no good animal model to study SPSD. A mouse model with anti-GAD65 T cells resembled the classic multiple sclerosis (MS) animal model, experimental autoimmune encephalomyelitis (EAE), rather than SPSD. Similarly, transfusion of GAD65 antibodies lead to certain motor and/or cognitive impairment in rats, but it failed to replicate the typical SPSD phenotype we observe in humans [48].

## 5. The Hen and the Egg: B Cells and T Cells in the Anti-GAD65 Immunopathophysiology

Antibodies against the synaptic antigen GAD65 are the most frequent biomarker of SPSD autoimmunity. GAD is directly involved in the synthesis of GABA as it is the rate-limiting enzyme for production. Apart from SPSD, a broad spectrum of GAD65-associated neurologic (e.g., refractory epilepsy, encephalitis, etc.) and non-neurological (e.g., T1DM, thyroid dysfunction, vitiligo as well as pernicious anemia) autoimmune diseases exist [13]. Antibodies targeting neuronal surface antigens, such as the glycine receptor, are considered pathogenic [49], but whether GAD65 antibodies are pathogenic or rather a marker of T cell-mediated autoimmunity is an ongoing debate. However, the level of GAD65 antibody titers and their changes during the disease course is not associated with disease severity or treatment response in SPSD [50,51,52]. 

While the antigen is intracellularly located (but close to the synapse, in contrast to the antigens of classic onconeuronal antibodies), the antibodies to GAD65 seem not to get internalized by neurons and hence should not be able to engage with the antigen [53]. Another explanation postulates the mediation of autoimmunity by T cells or by occurrence of other concomitant antibodies [54,55]. Conversely, work exists demonstrating epitope-specific effects on motor and cognitive functions by monoclonal antibodies to GAD65 in rats and impairment of GABAergic neurotransmission in vitro [48,56]. Epitope-mapping, however, has not revealed any differences with regard to GAD65 antibody-specificities across the various phenotypes [53,57]. A solid T cell response to GAD65 is suggested by the fact that GAD65 antibodies recognize linear epitopes and hence memory T cells could play a major part in the response against GAD65-expressing neurons, and the fact that infiltrating T cells are histologically proven in anti-GAD65 neurological syndromes [58,59]. However, evidence for an important role of B cell-mediated pathways is growing [41,43,58,60,61,62].

In autoimmune neurological syndromes associated with GAD65 antibodies (anti-GAD65 AINS), intrathecal synthesis of GAD65 antibodies is detected in 85–100% of patients but in none of the non-neurologic patients [41,43,63]. A marked intrathecal GAD65-specific IgG-synthesis indicates clonal B cell activation in the central nervous system [41,43]. There is evidence that GAD65 antibody-producing plasma cells can be recruited from memory B cells by activation via cytokines and toll-like receptor ligands and can also persist in survival niches in the bone marrow [61]. Even after autologous hematopoietic stem cell transplantation (aHSCT), GAD65 antibodies persisted in some SPSD patients, which may be a sign of inadequate accessibility of niches by this therapy [64]. Just recently, Biljecki and colleagues demonstrated the presence of intrathecal GAD65 antibody producing cells within the CSF compartment and an accumulation of GAD65-specific B cells and plasma cells in the CSF in early disease stages [62]. While T cell damage is believed to play a major part in SPSD pathophysiology and infiltrating T cells are histologically proven in anti-GAD65 AINS, B cell action appears to contribute significantly to the damage, even if not as main drivers but by possibly entertaining the T cell attack [58]. Hence, targeting B cells in the CSF compartment could be a promising treatment strategy.

## 6. Anti-GAD65-Associated Neurological Syndromes and Genetics

Information on genetic susceptibility or associations of certain gene variations to the different SPSD subtypes is scarce. The first report of the association of human leukocyte antigen (HLA)-DQB1*0201 with SPS dates back almost 30 years [65]. Since then, literature is almost exclusively limited to anti-GAD65 AINS. Clinical pictures of anti-GAD65 AINS show a broad spectrum of possibilities as well as a frequent occurrence of immunological co-phenomena. Recognition of different GAD epitopes as well as a diverse genetic background are discussed as causes [66]. In a multiplex family with anti-GAD65 AINS, an association with a rare human leukocyte antigen (HLA class II) haplotype (DRB1*15:01:01∼DQA1*01:02:01∼DQB1*05:02:01) was described in 2017 [67]. The work by Thaler and colleagues analyzed 33 genes associated with autoimmunity or representing immunological checkpoints and, additionally, HLA-DRB1 and HLA-DQB1 haplotypes in 19 patients with anti-GAD65 AINS [68]. Four of these patients had a diagnosis of SPS for which HLA haplotypes were explicitly reported (HLA-DRB1: 13:01, 13:01; 04:05, 07:01; 04:01, 13:02; 13:01, 13:02. HLA-DQB1: 06:03, 06:03; 02:02, 03:02; 02:01, 06:04; 06:03, 06:04). Within the four SPS patients, detected gene variants included cytotoxic T-lymphocyte-associated protein 4 (CTLA4) in all four patients. CTLA4 encodes an inhibitory immune checkpoint expressed on regulatory T cells, and the variants detected here have also been associated with T1D. One patient harbored a variant of unknown significance (VUS) in caspase-10 (CASP10); another patient harbored a VUS in syntaxin-binding protein 2 (STXBP2). Supplementary descriptions of an association of sporadic SPS to other HLA class II haplotypes already exist with the conclusion of a primary DQ effect on anti-GAD65 AINS [65,69]. Isolated case reports also describe an occurrence of SPS in HLA-B27-positive patients of non-neurological concomitant diseases (HLA B27 and spondyloarthropathy [70]; HLA B27 and Hodgkin’s lymphoma [71]). Recently, Strippel and colleagues performed a genome-wide association study (GWAS) and an associated analysis of the HLA region in a large German cohort of 167 patients with anti-GAD65 AINS, 48 of which with SPS, and 1047 control patients without neurological or endocrine disease [72]. The authors did not find any genome-wide significant associations to specific anti-GAD65 AINS, however, 400 associations at the nominal level of significance. Four genomic loci for GAD65-AINS were described as associated with SPS, in particular, the locus with the mapped gene Neogenin 1, and was classified as a relevant risk locus. All in all, this work could provide evidence for a genetic vulnerability to different anti-GAD65 AINS phenotypes. Future work is needed for a more accurate classification of GAD65 autoimmunity and genetic vulnerability. In addition, genetic work on non-GAD65 autoimmunity is essential for better understanding and classification of the complex pathophysiology of SPSD. 

## 7. Relevance of the Different Immunopathophysiologies for Therapeutic Strategies

Neuronal antibodies in SPSD, as in autoimmune movement disorders in general, can be categorized into three groups, based on the respective location of their antigen and their presumed pathogenic relevance [31]. The first group represents antibodies against neuronal surface antigens (e.g., DPPX, GlyR and GABA_A_R), which are assumed to have a direct pathogenic effect. A paraneoplastic occurrence is overall rare with neuronally mediated disorders. However, it is important to be aware of thymomas in anti-GlyR syndromes and B cell neoplasms with anti-DPPX. Generally, immunomodulatory therapies show promise in treating the diseases associated with neuronal antibodies [31,73,74]. 

The classical paraneoplastic or “onconeural” antibodies represent the second group (e.g., Ri and Zic4) and are directed against intracellularly localized antigens. As the effect is mainly mediated by cytotoxic T cells, the prognosis and response to immunotherapy is usually poor and the prognosis depends mainly on the underlying malignancy. 

The third group (e.g., GAD65 and amphiphysin) is, to a certain extent, intermediate to the first two groups, as the antibodies are directed against intracellular synaptic proteins. A paraneoplastic occurrence is typically seen with amphiphysin antibodies (breast and lung cancer), and rare with GAD65 antibodies. Overall, this group also shows a moderate response to immunomodulatory therapy. As mentioned above, whether these antibodies play a pathogenic role is still unclear and an ongoing topic of investigation.

## 8. Immunomodulatory Therapies—State of the Art and Future Outlook

Pharmacological treatment of SPSD is multifaceted and includes symptomatic treatment, immunotherapy and, if necessary, tumor treatment, and requires a continuous monitoring of potential side effects and disease activity. Symptomatic therapy generally consists of GABA-enhancing drugs. Immunomodulatory therapy, primarily IVIg, may be started, if symptomatic treatment is insufficient or can be started overlapping. Pragmatic management of SPSD treatment has been previously proposed [75]. The success of therapy is strongly linked to the underlying pathophysiology. For example, a positive treatment response may be limited when relevant tissue damage by cytotoxic T cells has already occurred. In this section, we will refer to current knowledge on immunotherapy (summarized in Table 1), provide an outlook, and highlight what we can learn from other autoimmune neurological diseases.

The first controlled study on immunotherapy in SPSD dates back to 2001, which described high-dose IVIg as an effective therapy against motor symptoms in anti-GAD65- associated SPS. The duration of the effect varied between patients from a few weeks to a year [76]. A follow-up study on the long-term course of 36 patients with IVIg maintenance therapy every 3–6 weeks was recently published [77]. Response to IVIg was measured by patients’ subjective treatment response, physician-observed effects, modified Rankin Scale and different dependency trials. In two thirds of the patients, a clinically meaningful response over a median period of 40 months was observed, but in almost one third of patients, treatment benefit declined over a similar time period. Of note, patients who did not respond within the first 3 months of administration remained unresponsive, even with IVIg continuation for several months. To date, IVIg is considered first-line therapy, although there is no universal dosing regimen and the therapy is adjusted at an individual level based on the patient’s tolerability, treatment response and/or adverse side effects (e.g., risk of thrombosis, renal dysfunction and/or aseptic meningitis). Subcutaneous immunoglobulin (SCIG) has also recently become available as an alternative treatment in AINS and has been studied in an SPSD patient population that did not tolerate IVIG, and showed good overall tolerability along with the persistence of a treatment effect [78]. Administration for some patients is easier and the systemic side effect profile appears lower. Injection site reactions are common with SCIG but usually well tolerated. Also, achieving the desired dose equivalent to IVIG is more difficult with this method. Further long-term data and controlled studies are currently lacking.

New on the horizon of therapies for AINS that impact IgG pathways are neonatal Fc receptor (FcRn)-targeted therapies [79]. Primarily expressed in endothelial and myeloid cells, FcRn promotes IgG recycling and prolongs the life of IgG molecules. Blockade of FcRn prevents endogenous IgG from binding the receptor, leading to IgG depletion by lysosomal degradation. This could be exploited therapeutically by limiting the lifetime of pathological autoantibodies as well as by prolonging the lifetime of therapeutic antibodies. In EAE models, FcRn-targeted therapies caused degradation of overall IgG [80]. By appropriate preparation in the laboratory, it was shown that this depletion can also be prepared in an antigen-specific manner to affect total IgG levels less severely [81]. The experiments resulted in a significant reduction of disease activity. FcRn-targeted therapies are currently being tested in various autoimmune neurological diseases and available data are limited. 

As in other autoimmune neurological diseases, steroid therapy, usually as pulse therapy followed by tapering and oral maintenance dose, is easily available and can play a role for short-term care. It has long been known that steroids should rather be used as a relapse therapy due to their long-term side effects, but there are varying reports on the degree of efficacy in SPSD [5,75]. In particular, patients with GAD65-associated SPSD may frequently suffer from concomitant T1DM, which should prompt caution using steroid therapy, as it has a direct impact on blood glucose levels. 

Plasmapheresis and plasma exchange (PLEX) have been shown to be an alternative to steroid therapy in acute relapses within autoimmune neurological disorders. While there are no randomized trials, there is evidence of benefit in SPSD [82]. A recently published work reported (temporary) improvement of symptoms in 59–75% of cases [83]. Moreover, over 50% of the patients studied were able to reduce GABAergic symptomatic medications. Only 10% of patients experienced a PLEX-related adverse event, none of which were classified as serious. Due to administration and potential hemodynamic effects, it is used most often for acute exacerbations.

As an alternative, or in cases of a refractory disease course, anti-CD20 B cell-depleting therapies can be considered. Rituximab has been used for decades to treat rheumatological conditions and demonstrated very good efficacy in autoimmune neurological diseases such as MS, neuromyelitis optica spectrum disorders (NMOSD), myasthenia gravis and various autoimmune encephalitides. As B cells are thought to play at least a partial role in the pathophysiology of SPSD, rituximab has also been tried in this setting. Depletion of antigen-specific reactive memory B cells may prevent long-term production of potentially pathogenic antibodies. Moreover, there is less T cell activation when there are decreased numbers of circulating B cells and this downstream effect is likely part of the efficacy of B cell therapies in autoimmune diseases. However, the total number of studies and SPSD patients receiving rituximab is small. The largest controlled trial conducted in SPSD reported no significant changes of stiffness, quality of life scores and heightened sensitivity after 6 months [84], but there are reports that patients can benefit from rituximab therapy especially when treated for a longer period of time. This may be even more so if patients are younger, therapy is initiated early, and in patients who are less disabled at the start of therapy or if they already had a subtherapeutic response to other immune therapies [84,85]. If pathogenic antibodies are involved or there is an inadequate effect or intolerance to IVIg, rituximab should be up for consideration.

New on the horizon is the use of bortezomib, a proteasome inhibitor known for administration in multiple myeloma, which depletes plasmablasts rather than memory B cells. One case report noted improvement of mobility and reduction of symptomatic therapy in a patient with an SPSD refractory to IVIG, mycophenolate mofetil and rituximab [86]. Some experience exists in other autoimmune encephalitides and, at the current time, clinical trials are ongoing in various inflammatory neurological diseases [87]. Even less experience exists with CD38 receptor-specific therapy (i.e., daratumumab), which is designed to target CD20-negative long-lived plasma cells and has been effective in severe cases of autoimmune encephalitides in single case reports [88,89]. 

A different class of medications, which have had some therapeutic benefit in various CNS inflammatory diseases (e.g., NMOSD and myelin oligodendrocyte glycoprotein antibody-associated disease [90]) but for which there are no data regarding the effect in SPSD, are anti-interleukin-6-receptor antagonists such as tocilizumab. Whether these therapies will also play a role in SPSD treatment remains to be seen.

As an ultima ratio, small numbers of SPSD patients who received aHSCT also exist. The most common neurological disease in which aHSCT has been used as a therapeutic option is MS [91] with the main goal of treatment to restore immunologic tolerance by aggressive depletion of the immune system. Sanders and colleagues were the first to report a good and long-lasting effect in two female patients with GAD65 antibody-associated SPSD without unexpected toxic effects [92]. Over time, more refractory SPSD cases treated with aHSCT have been reported with positive results, whereas not all of them were associated to GAD65 antibodies [93,94]. However, a prospective single-arm study was less promising and terminated prematurely due to the lack of treatment effect [95]. Of the responders, less than half were considered to be in remission a few years after the treatment intervention. Moreover, the majority of patients required ongoing immune and symptomatic therapies after aHSCT [96]. AHSCT could be an effective treatment in the right patient population, but a variety of potentially serious adverse events (in particular, infections) as well as immunologic co-phenomena can occur. Interestingly, SPSD can also occur as a post-transplant phenomenon after aHSCT for other diseases [97].

Little is known about which patients with an associated antibody benefit from immunotherapy, when the ideal time to start therapy is, and how long immunotherapy should be continued. Additionally, there is an unmet need for biomarkers that can monitor disease activity and effectiveness of immunomodulatory therapies. In any case, the category of neuronal antibody should play a crucial role in these considerations. We know from autoimmune diseases such as MS and NMOSD that therapy should be started as early as possible to avoid future disability [98,99]. Data in SPSD in this regard are lacking, but it appears that there may be subgroups of patients that benefit from starting immune therapies early [31,51]. Given the disease burden and disability that can come from SPSD, it seems prudent to consider immune therapies earlier rather than later in the disease course, and to critically reevaluate the therapy on a regular basis. 

## 9. Special Treatment Considerations: Pregnancy with SPSD and SPSD in Childhood

As women are much more frequently affected by SPSD than men, the role of pregnancy both for the clinical course as well as treatment consideration merits particular mention. Data on the clinical course of SPSD during pregnancy are sparse and mainly based on single case reports. 

In the second and third trimester of pregnancy, there is an estradiol-driven cytokine shift due to a decreased production of inflammatory Th1 cells and for this, increased production of anti-inflammatory Th2 and Th17 cells. This leads to a reduction of proinflammatory cytokines and an increase of anti-inflammatory cytokines, probably to protect the growing fetus [100]. In MS, Th1 cells are thought to play an important role and the mother is in a relatively protected state during pregnancy, but has a higher risk of relapsing disease activity postpartum [101]. An opposite example is shown by pregnant women with NMOSD, where pregnancy is associated with an increased risk of disease activity and higher disability postpartum [102]. However, literature on pregnancy in SPSD is scarce. Recently, nine pregnancies in six patients were summarized in a systematic review [103]. At the very least, improvement in symptom intensity during pregnancy appears to be possible, with significant reductions in symptomatic benzodiazepine therapy reported in over 50% (5/9) of pregnancies. In all cases, symptoms worsened postpartum and symptomatic therapies had to be restarted or even increased in dosage. Overall, healthy, unaffected babies were born in all reported cases. Interestingly, two of the newborns showed asymptomatic high titers of GAD65 antibodies in serum.

While symptomatic therapy with benzodiazepines is in principle not contraindicated in pregnancy, GABA enhancers come with the risk of neonatal dependence. IVIg can be used in early pregnancy and steroids or PLEX can also be used, but should be carefully considered at the individual level alongside with a maternal fetal medicine specialist. Most information about the use of B cell-depleting agents and pregnancy in neurologic disorders originates from MS. Studies in MS have shown that B cell therapies are effective for prevention of inflammatory activity before pregnancy and in the postpartum period [104]. Starting de novo during pregnancy is usually not recommended, although no clustered birth complications associated with B cell-depleting agents have been reported to date. 

An interesting but also problematic area is pediatric-onset SPSD. Due to the rare occurrence of the disease in childhood and limited knowledge of specific pediatric features, it is not uncommon for diagnosis to be delayed until adulthood, by which time relevant disability may have already occurred [105]. Yeshokumar and colleagues reported that, despite a similar clinical phenotype to adult-onset SPSD, nearly two thirds of pediatric SPSD patients were not diagnosed until adulthood. It is not difficult to imagine that the disease leads to significant limitations in the social and daily life of children. Consequently, pediatricians need to be aware of the clinical features of SPSD to initiate early referrals to specialists who can implement therapies in hopes to prevent future disability.

## 10. Summary

Recent research has fueled a wider understanding of SPSD. The most important clinical developments relate to the recognition of a wider affection, beyond the classic core motor symptoms. The progress with regard to immunomodulatory treatment seems promising, especially if considering the new agents being tested in other autoimmune neurological diseases. Crucial knowledge gaps still exist with regard to the exact immunopathophysiology of SPSDF (e.g., the role of T and B cells, autoantibodies, etc.) and what might be the best treatment target(s) for SPSD.

## Figures and Tables

**Figure 1 biomedicines-11-02500-f001:**
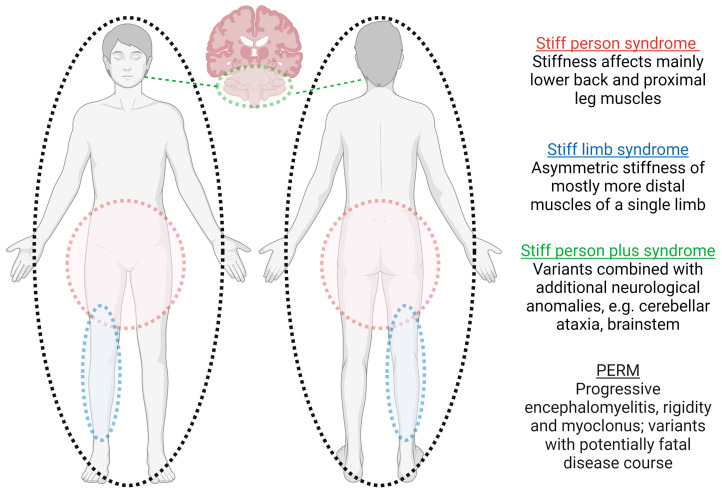
Clinical phenotypes of SPSD. Created with BioRender.com (accessed on 31 July 2023).

**Figure 2 biomedicines-11-02500-f002:**
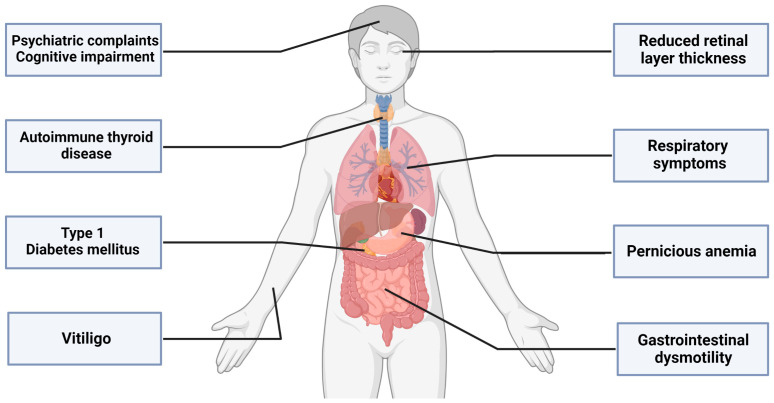
Features in SPSD beyond the classic motor symptoms. Created with BioRender.com (accessed on 31 July 2023).

**Table 1 biomedicines-11-02500-t001:** Summary of immunomodulatory therapy aspects in SPSD.

IVIg is considered the first-line treatment where available Corticosteroids and plasmapheresis are other options of first-line treatment In spite of a negative trial, Rituximab is useful in some cases aHSCT may be a treatment option in treatment refractory cases FcRn-targeted therapies, interleukin-6-receptor antagonists, plasmablast- and plasma cell-targeting therapies and the proteasome inhibitor bortezomib are used in other neuroimmunological diseases and might be possible treatment options in the future There is an unmet need for biomarkers to evaluate treatment response Further studies are needed to determine the best approaches to treatment initiation, escalation and maintenance

## Data Availability

Not applicable.

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
