# Peer review of "Stiff Person Spectrum Disorders—An Update and Outlook on Clinical, Pathophysiological and Treatment Perspectives"

_biomedicines, 2023, doi:10.3390/biomedicines11092500_

Round 1
Reviewer 1 Report
Balint and colleagues provide an update on clinical manifestations of Stiff Person Spectrum Disorders (SPSD) with the current understanding of immunopthophysiological mechanisms and therapeutic strategies.
This is a comprehensive and well-written review with substantive and practical value. It expands on non-motor manifestations of the disease and nicely summarizes the current and future therapeutic consideration and unmet needs. Special treatment considerations for SPSD in childhood and during pregnancy is an important topic as it relates to a better recognition of challenges and complexities involved in diagnosing and treating patients along the continuum of the disease.
This review paper stands out by its inclusiveness and clarity. It is an unbiased up-to-date account of Stiff Person Spectrum Disorders, useful not only for practicing neurologists but also relevant to a broader readership.
Author Response
Response: We thank the reviewer for the very positive feedback!
Reviewer 2 Report
The article is consistent within itself. The references are relevant and recent. The cited sources are referenced correctly. Appropriate and key studies are included. The paper is comprehensive, the flow is logical and the data is presented critically.
However, there are some specific comments on weaknesses of the article and what could be improved:
Major points - none
Minor points - most of the critical points are for the structure and headings of the paper.
1. Possible cellular immune mechanisms are superficially mentioned (a couple of sentences would be enough)
2. Before starting the recent and novel therapies, please discuss briefly the classical treatment with focus on the lack of effectiveness, etc.
3. Section 3 - There is a repetion, please, revise ". The diagnosis – about swallows and summers, antibodies and diagnosis"
4. "The hen and the egg, B cells and T cells in the anti-GAD immunopathophysiology" - I would suggest to use colon "The hen and the egg: B cells and T cells in the anti-GAD immunopathophysiology"
5. "6. GAD65 neurological syndromes and genetics" - "6. GAD65-associated neurological syndromes and genetics
The English grammar and style is good.
Author Response
We thank the reviewer for the thoughtful review of our work.
Minor points - most of the critical points are for the structure and headings of the paper.
- Possible cellular immune mechanisms are superficially mentioned (a couple of sentences would be enough)
- Response: We revised the section 5 and added greater detail to cellular immune mechanisms.
- Before starting the recent and novel therapies, please discuss briefly the classical treatment with focus on the lack of effectiveness, etc.
- Response: We revised section 8 and added new aspects and discussion of the classical treatment.
- Section 3 - There is a repetion, please, revise ". The diagnosis – about swallows and summers, antibodies and diagnosis"
- Response: We thank you for pointing out the repetition. We changed the headline to “The diagnosis – about antibodies, swallows, and summers ”
- "The hen and the egg, B cells and T cells in the anti-GAD immunopathophysiology" - I would suggest to use colon "The hen and the egg: B cells and T cells in the anti-GAD immunopathophysiology"
- Response: We agree and changed the headline accordingly.
- "6. GAD65 neurological syndromes and genetics" - "6. GAD65-associated neurological syndromes and genetics
- Response: Again, we agree and decided to change the headline to “Anti-GAD65 associated neurological syndromes and genetics"